# Characterization of Oligofructose-Induced Acute Rumen Lactic Acidosis and the Appearance of Laminitis in Zebu Cattle

**DOI:** 10.3390/ani10030429

**Published:** 2020-03-04

**Authors:** Rejane dos Santos Sousa, Francisco Leonardo Costa de Oliveira, Mailson Rennan Borges Dias, Natalia Sato Minami, Leonardo do Amaral, Juliana Aparecida Alves dos Santos, Raimundo Alves Barrêto Júnior, Antonio Humberto Hamad Minervino, Enrico Lippi Ortolani

**Affiliations:** 1Department of Clinical Science, College of Veterinary Medicine and Animal Science, University of Sao Paulo (FMVZ/USP). Av. Prof. Orlando Marques de Paiva, 87, Cidade Universitária, São Paulo CEP 05508-270, SP, Brazil; rejane.santossousa@gmail.com (R.d.S.S.); oliveiraflc@usp.br (F.L.C.d.O.); mailsonveterinario@gmail.com (M.R.B.D.); minaminatalia@usp.br (N.S.M.); leonardoamaral38@hotmail.com (L.d.A.); jhu_unesp@yahoo.com.br (J.A.A.d.S.); 2Department of Animal Science, Federal Rural University of the Semiarid Region (UFERSA). Av. Francisco Mota, s/nº—Bairro Pres. Costa e Silva, Mossoró CEP 59625-900, RN, Brazil; barreto@ufersa.edu.br; 3Laboratory of Animal Health (LARSANA), Federal University of Western Pará (UFOPA) Rua Vera Paz, s/n, Salé, Santarém CEP 68040-255, PA, Brazil

**Keywords:** nelore, ruminal acidosis, symptoms, induction model

## Abstract

**Simple Summary:**

In this paper we characterize oligofructose-induced acute rumen lactic acidosis in zebu cattle focusing on alterations of the experimental protocol concerning ruminal condition. Here we describe zebu cattle’s lack of adaptation to the oligofructose induction model for laminitis and we provide methodology of adaptation that allows the higher occurrence of laminitis with a lower degree of ruminal acidosis and better animal welfare. Additionally, we compare the animals that required and did not required supportive treatment, to try to understand which factors are related to susceptibility. The induction model promoted marked reduction in rumen pH, rumen anaerobiosis, carbon dioxide pressure, and an increase in rumen lactate, blood osmolarity, and cortisol concentration. The animals treated had lower values of rumen pH and marked dehydration, evidenced by the increase in globular volume and serum urea. The clinical condition caused by excess oligofructose is more severe than the classical sucrose induction model, but is efficient in producing laminitis.

**Abstract:**

The objective of this study was to characterize oligofructose-induced acute rumen lactic acidosis and its consequences in zebu cattle. We used 29 Nellore heifers which were submitted to experimental induction of laminitis by oligofructose excess. During the induction period, the animals underwent clinical examination, including laminitis diagnosis (hoof pressure testing and locomotion score) and blood and ruminal fluid sampling every six hours (over the initial 24 h) and every 12 h (up to 72 h), after the highest dose. Almost half of the animals (48.1%) required treatment with bicarbonate and saline to correct metabolic acidosis and dehydration. Due to this treatment, the animals were analyzed in treated (n = 13) and non-treated (n = 14) groups. The induction model promoted marked reduction in rumen pH, rumen anaerobiosis, carbon dioxide pressure, and increase in rumen lactate, blood osmolarity, and cortisol concentration. The animals treated had lower values of rumen pH and marked dehydration, evidenced by the increase in globular volume and serum urea. The clinical condition caused by excess oligofructose is severe, with the differential of the appearance of ephemeral fever and respiratory compensation against systemic acidosis, in addition to the frequent appearance of laminitis.

## 1. Introduction

The rumen can be considered a symbiotic environment, consisting of bacteria, protozoa, and fungi, which provide ruminants with the ability to digest soluble and complex carbohydrates such as cellulose and hemicellulose, and in return receive a nutrient-rich environment and favorable conditions for development [1,2]. These microorganisms are adapted to an anaerobic environment and have a ruminal pH range between 5.7–6.5, however, this harmonic symbiosis can be modified due to changes that promote changes in rumen pH, leading to an overlap of one microorganism population over another [3].

Among the changes in rumen pH values, acute rumen acidosis (ARA) is caused by the abrupt consumption of a large amount of soluble carbohydrate, causing a sudden reduction in rumen pH to values below five, favoring the growth of gram-positive bacteria such as *Streptococcus bovis*, which produce lactic acid [4,5,6]. When the ruminal pH drops below 4.5, the *Streptococcus bovis* population dies, and the environment favors the population growth of Lactobacillus that produce more lactate [7,8,9].

Lactate production may culminate in the occurrence of metabolic acidosis and ruminal wall lesions, causing ruminitis. The abrupt reduction in rumen pH also promotes the death of gram-negative bacteria, releasing lipopolysaccharides (LPS) that are endotoxins from the cell wall [10]. LPS has been implicated as a trigger in the development of several inflammatory processes such as laminitis [10,11,12]. Metabolic acidosis may cause oxidative stress, as mitochondria are stimulated to increase reactive oxygen species (ROS) production and to decrease adenosine triphosphate (ATP) production, which may further aggravate the condition of the animal [13,14,15].

Laminitis is considered a systemic disease with manifestation in the digits, in which there are vascular and degenerative changes of the laminar chorion, which can affect the positioning and mobility of the digital phalanx within the horn case [16]. ARA is still considered the main triggering factor for laminitis due to the role of endotoxins, but its pathophysiology is not well known [17,18].

In the microcirculation of the digits, these endotoxins are believed to promote vasoconstriction, with increased capillary pressure, thrombus formation and endothelial injury, leading to fluid and blood cell leakage in the various regions of the digital chorion, with increased pressure within the hoof and inadequate oxygenation of the dermis and epidermis [16].

The laminitis induction model developed by Thoefner et al. [19] using oligofructose overload in taurine cattle has contributed to the progress of better understanding of rumen lactic acidosis and the pathophysiology of laminitis, but information on this type of induction in zebu cattle could not be found in the literature. Nevertheless, previous reports with experimental acidosis induction using the sucrose model suggest that zebu cattle are more resistant than taurine to lactic acidosis [9,20]. Thus, the present work aims to characterize the condition of oligofructose-induced ARA, and to evaluate the appearance of laminitis in zebu cattle.

## 2. Materials and Methods

### 2.1. Animals and Feed

A total of 29 three-year-old Nelore heifers, weighing 474.5 ± 58.5 kg, were used. The animals were dewormed and vaccinated against clostridiosis; they underwent surgery for ruminal cannula implantation, which was followed by a 30-day recovery period. The animals received a basal diet calculated at 2.3% of live weight, composed of 60% dry matter of Coast-cross grass hay and 40% concentrate with 14% of crude protein, offered once daily. The cattle were provided 60 g of a commercial mineral supplement daily and had free access to water. This study was approved by the Animal Ethics Commission of the College of Veterinary Medicine and Animal Science at the University of São Paulo.

### 2.2. Pilot Protocol

Thoefner et al. [19] developed an ARA induction protocol and consequent laminitis for taurine cattle using the 17 g/kg dose of oligofructose. In this study, a pilot test was performed in zebu cattle using 17 g/kg oligofructose, but the animals presented severe ARA and the induction was interrupted before the onset of laminitis. Subsequently, a second pilot test was performed with a reduction of 10% of the recommended dose, using 15.3 g/kg; this dose was efficient in causing laminitis and ARA was less severe.

### 2.3. Study Design

The study had a completely randomized design, in which the animals were initially subjected to induction of ruminal lactic acidosis (ARA) using oligofructose, according to a methodology adapted from Thoefner et al. [19], that is, 15.3 g/kg of oligofructose (Orafti® P95, Beneo GmbH, Mannheim, Germany) was divided into one main dose of 10.71 g/kg and adaptation doses of 1.53 g/kg for three consecutive days.

The animals were evaluated through hoof pressure testing and filmed locomotion score at the following times: 72, 48, and 24 h before induction (T-72, T-48, and T-24), when the animals received the adaptation doses of 1.53 g/kg oligofructose per day, administered as a dose of 0.765 g/kg in the morning and afternoon. The moment immediately prior to administration of 10.71 g/kg of oligofructose (the main dose) was considered to be the baseline (T0), after which the animals were evaluated every 6 h for 24 h (T6, T12, T18, and T24), and then every 12 h for an additional two days (T36, T48, T60, and T72) to check for the appearance of laminitis. Oligofructose was diluted in warm water and added to the rumen via the ruminal cannula. During the entire study, the diet (concentrate and hay) was freely available to the animals.

### 2.4. Clinical Examination and Ruminal Fluid Sampling

At the aforementioned moments (T-72 to T72) physical examination of the animals was performed according to classical recommendations [21], with the measurement of heart rate, respiratory rate, ruminal movement and the evaluation of the presence of clinical signs such as enophthalmia, dry muffle, nasal discharge, ocular discharge, ataxia, tremors, left abdominal flank distension, and signs of hypovolemia, such as decreased extremity temperature. The ruminal mucosa was visually inspected through the rumen cannula evaluating for the presence of swollen areas or loss of keratinized epidermal layer, indicative of ruminitis.

During clinical evaluation ruminal fluid samples were obtained to evaluate the following variables: pH, oxidoreduction potential (ORP), methylene blue reduction time (MBRT) titratable acidity (TA), lactate L (levogyre), and osmolarity. Ruminal fluid samples were collected directly from tree regions of the rumen (cranial, medial and caudal). The contents were quickly filtered with gaze and immediately analyzed for ruminal pH, ORP, MBRT and TA. Subsamples were stored in plastic tubes at −20 °C for the determination of lactate L and osmolarity.

Analysis of pH and ORP were performed using a benchtop pH meter (DM-22, Digimed, São Paulo, Brazil), while titratable acidity and methylene blue reduction test were performed by classical methods [17].

Rumen L-lactate concentration was determined by a commercial kit (Randox^®^, County Antrim, UK) on an automated biochemical analyzer (RX Daytona, Randox^®^, County Antrim, UK) and osmolarity on a freezing point osmometer (The Advanced Micro Osmometer 3300, Advanced^TM^, Norwood, MA, USA).

### 2.5. Blood Sampling and Analysis

Blood samples obtained with of vacuum tubes without anticoagulant were kept at room temperature until clot formation and centrifuged at 1000× *g* for 15 min to obtain serum, which was aliquoted into microtubes and frozen at −20 °C. Serum samples were used for determination of cortisol, lactate L, serum osmolarity, biochemistry (urea and creatinine) concentrations, and determination of sodium (Na^+^), potassium (K^+^), and chlorine (Cl^−^) electrolytes. Electrolytes were determined on a Cobas B 121 hemogasometer (Roche Diagnostics, Basel, Switzerland). The anion gap was calculated according to Kaneko et al. [22] = (Sodium + Potassium) − (Chlorine + Bicarbonate).

Samples collected with diethylenediamine tetraacetic acid (EDTA), and with EDTA and sodium fluoride, were promptly refrigerated at 4 °C. EDTA samples were used to perform a blood count on a BC-2800 Vet hematological counter (Mindray^®^, Shenzhen, China) for globular volume (hematocrit), red and white blood cell counts, and hemoglobin concentration. Samples obtained with fluorine were centrifuged for plasma which was used to determine L-lactic acid concentrations. Plasma L-lactate and serum urea and creatinine concentrations were determined using commercial Randox^®^ kits on an RX Daytona (Randox Laboratories Ltd, Crumlin, UK) automated biochemical analyzer.

For blood gas analysis, arterial whole blood was collected from the auricular artery in 3 mL syringes containing sodium heparin. Blood pH, carbon dioxide partial pressure (pCO_2_), oxygen partial pressure (pO_2_), bicarbonate concentration (${\mathrm{HCO}}_{3}^{-}$), and base excess (BE) values were obtained using an iStat gas analyzer and ECG8+ cartridges (Abbott^®^, Chicago, IL, USA).

### 2.6. Laminitis Clinical Diagnostic

The gold standard for determining the occurrence of laminitis was the hoof sensitivity test (hoof pressure testing), considering the occurrence of laminitis when the animal had two positive responses to the hoof sensitivity test. The animals also underwent filmed locomotion score evaluation to determine the onset of lameness. Both tests were performed during all periods mentioned above. In this study, laminitis was defined as a clinical disease with the manifestation of sensibility and/or detectable lameness, identified shortly after fermentable carbohydrate overload [19].

For the hoof sensitivity test, the forelegs were suspended and a force was applied to each digit in different areas of the hoof. The pressure exerted was only to visually assess whether the animal responded with muscle fasciculations or attempted limb removal. The reaction was subjectively classified as either 0, without reaction to the pressure exerted, or 1, as the presence of reaction.

### 2.7. Histopathological Confirmation of Laminitis

Between 96 h and 120 h after oligofructose induction, the animals were submitted to hoof biopsy. For this, the animals were contained in lateral decubitus, anesthetized by the technique of Bier and examined for hoof sensitivity. The digit that presented pain response was submitted to biopsy. The biopsy was performed on the medial third of the dorsal surface of the hoof initially wearing down of the stratum corneum using a micro-rectifier-typerotary tool (Dremel 4000^®^, Robert Bosch Tool Corporation, Mt. Prospect, IL, USA) until the soft tissue was noticeable through palpation with a hypodermic needle. Afterwards, a 7 mm punch was used to remove the tissue fragment, which was stored in 10% formalin solution [23]. The tissue fragment was submitted to the routine histological technique and stained with eosin and hematoxylin.

### 2.8. Supportive Treatment

After ARA induction, animals that had a blood pH below 7.2 at T12 or T18 were treated with 1ml/kg sodium bicarbonate solution (8.4%) and 6 L of isotonic saline solution (0.9% NaCl) [24]. Six hours after the first treatment, if the blood pH did not increase, the treatment was repeated.

### 2.9. Statistical Analysis

Statistical analyzes were processed with the aid of a statistical analysis computer program, the statistical analysis system (SAS). The data obtained during the experimental period were analyzed for their distribution using the Kolmogorov–Smirnov test and the homogeneity of the variances was evaluated. Although the same dose of oligofructose was used to induce rumen acidosis, some animals required supportive treatment and some did not. Thus, the variables studied were compared in treated (T) and non-treated (NT) animals. The characterization data of rumen lactic acidosis and laminitis were evaluated by analysis of variance by the PROC ANOVA procedure and the mean of each time was compared using the Bonferroni average test. To compare treated and non-treated animals, analysis of variance was performed using the Bonferroni test for comparison between times.

In the study of the relationship between any two variables, the Pearson correlation coefficient (*r*) and the determination coefficient (*r*^2^) were calculated using PROC CORR. It was established that there was a high intensity correlation between the variables. The significance level adopted in this work was *p* < 0.05.

## 3. Results

### 3.1. Pilot Study

In order to develop the ARA experimental framework, two initial pilots were performed to verify the degree of ruminal acidosis obtained, using the protocol proposed by Thoefner et al. [19]. The first time, following the recommended doses of oligofructose of 17 g/kg, the ruminal pH reached incredibly low values (3.57) at T18 accompanied by decreased blood pH (7.14). This condition generated a disturbing clinical condition characterized by severe diarrhea, marked dehydration, sternal decubitus and major depression in the general condition. Such a set of symptoms required treatment that prevented the evolution of the probably fatal condition, necessitating the trial be aborted; failing to follow the appearance of laminitis.

In view of the severity of the condition, the main dose was reduced by 10% (from 7.65 g to 6.855 g archiving rumen pH of 4.3 and blood pH of 7.28 at T18 without jeopardized animal health and with no need of treatment. In addition, this new dose caused, at T24, an increase in the sensitivity of one digit (right medial) which then progressed to all digits of the forelimbs. Due to the lower severity of ARA obtained in the second pilot, the same oligofructose dose reduction was maintained until the end of the experiment, achieving 93.1% success in inducing laminitis in the animals used (27/29). From the 29 cows, a total of 13 (44.8%) needed supportive treatment at T12 or T18, and of those 46.15% (6 of 13) required a second treatment. The supportive treatment had no effect on the signs of laminitis in all animals treated.

### 3.2. Characterization of Oligofructose-Induced ARA

Besides the successful induction of acidosis in all 29 animals, two of them (6.89%) showed no sensitivity in the hoof and were removed from the experiment. To better understand the development of metabolic acidosis and the major depression in the general state that required the need for supportive medication, some variables were compared between treated (n = 13) and non-treated (n = 14) animals. T12 was the moment when the animals presented major ruminal and blood changes, so we compared these moments in the treated and non-treated groups to try to understand this process, since not all animals needed supportive treatment.

Initially, ruminal pH was compared at T12, with no significant difference in this variable (*p* = 0.143), and with the following pH values: treated 3.98 ± 0.13; untreated 4.07 ± 0.16. However, when comparing blood pH at the same time, there was a significant difference (*p* < 0.001) between the groups: treated 7.13 ± 0.04 and non-treated 7.26 ± 0.04. Interestingly there was a very high coefficient of determination (*r*^2^ = 0.842; *p* = 0.004) between rumen pH and blood pH only in the treated group but not in the untreated (*r*^2^ = 0.088; *p* = 0.294).

We also analyzed the difference in blood osmolarity at T12 and at baseline; higher mean differences (*p* = 0.002) were obtained in the treated group (46.6 ± 27.8 mOsm/L) than in the non-treated group (15.6 ± 10.3 mOsm/L). In both groups, blood osmolarity and pH were correlated. While in the treated group there was a significative inverse correlation (*r* = −0.692; P = 0.009; *r*^2^ = 0.48), in the non-treated group this fact did not occur (*r* = 0.003; *p* = 0.967). Plasma volume deficit was also compared, with a higher mean value (*p* = 0.015) for treated (36.3 ± 11.1%) than untreated animals (21.7 ± 15.6%).

### 3.3. Ruminal and Blood Variables

In order to better understand the ruminal changes against the new corrected dose of oligofructose, the results for the following variables are presented below: pH, oxide-reduction potential (ORP), titratable acidity (TA), methylene blue reduction time (MBRT), lactate-L, and osmolarity. There were no differences in all variables from T-72 to T0. At T6, lower ruminal pH values were found, with higher ORP, MBRT, and TA (Table 1). Higher plasma L-lactate levels were found at T24 than at T0, T6, and T-24 (*p* = 0.007). Osmolarity was higher at time T6 than at times between T-72 and T0 (*p* = 0.040). Mean cortisol concentrations were higher at T18 and T24 than at T12 (*p* < 0.001).

Higher positive correlation coefficients were observed between osmolarity and TA (*r* = 0.657; *p* < 0.001), osmolarity and MBRT (*r* = 0.696; *p* < 0.001), ORP and TA (*r* = 0.65718; *p* = 0.0001) and ORP and MBRT (*r* = 0.696; *p* < 0.001). While larger negative associations between two variables were detected between pH and ORP (*r* = −0.975; *p* = 0.001) and pH and TA (*r* = −0.666; *p* < 0.001). Blood variables, L-lactate, osmolarity, and cortisol did not differ between the untreated and treated groups.

Comparisons within the NT and T groups of the hematocrit and red blood cell (RBC) variables identified higher values at T6, T12, and T18 compared to all other moments studied (*p* = 0.031). Comparison of hemoglobin data within the NT group identified higher levels at T18 and T24 compared to all other times (*p* = 0.017), while in group T the T18 was higher than the others (*p* = 0.042), except for T24. When comparing the groups of these three variables (RBC, hematocrit and hemoglobin), higher values were found at T6, T12, and T18 than at the other times in the NT group (*p* = 0.027). The number of leukocytes (WBC) within group T was lower at T24 than at all times except T18 (*p* = 0.036), there were no differences within group NT (*p* = 0.702). PVD was more negative within the NT group at T18 and T24 (*p* = 0.002), the same occurring at T18 in relation to T12 and T6 in group T (*p* = 0.02) (Table 2).

Higher rumen lactate levels were detected in the rumen from T6 in untreated and treated groups, reaching higher values in T12 and T18 (*p* = 0.037); at T24 there was a reduction, but did not differ from T6. There was no difference between the groups regarding this variable (Table 3). Higher ruminal osmolarity was detected in T6 in group NT, in later times there was a reduction, while in group T there was no change for this variable over time. A comparison between groups showed higher ruminal osmolarity in group NT than in T in T6, while in T18 greater value was found for group T (Table 3). Urea values within the NT group were higher at time T-72 compared to times T6, T12 and T18 (*p* = 0.043). Within group T the T-72 was higher than the T12 and T18 times (*p* = 0.026). Among the groups, urea concentrations were higher in group T at times T6 and T24 (*p* = 0.049). Regarding creatinine, there was no difference between the times within the NT group (*p* = 0.999), but a higher value was found in T24 compared to the times T-72 to T6 (*p* = 0.026) in the T group. The variable was higher in group T at time T24 compared to group NT (*p* = 0.009).

Lower blood pH values were observed from T12 to T24 compared to all other times, within the NT group (*p* = 0.005). The same lower pH results were found at T12 and T18 compared to T24 within group T (*p* = 0.016). Comparing the groups, it was confirmed that the blood pH was lower in group T from T6 to T18 during induction (*p* = 0.024) (Table 3). Similar behavior in the results was found in the concentrations of bicarbonate and base excess (BE) both within and between groups. Thus, within both groups the levels of bicarbonate and BE at T12, T18, and T24 were lower compared to the other moments (*p* < 0.001). The animals treated had lower values of these variables above at T12, T18, and T24 compared to the NT group (*p* < 0.001) (Table 3).

There was a drop in blood CO_2_ pressure in both groups T and NT from T12 to T24, compared to the other times (*p* < 0.001). There was no difference in these values at the various times studied between groups T and NT (*p* = 0.691). There were no differences within or between the groups regarding sodium (*p* = 0.740) and potassium (0.978) values. Within the NT group only mean chlorine values at T6 were higher than T24 (*p* = 0.030), but there were no differences in group T. The anion gap was higher within the NT group at time T18 than at all other times except T12 and T24, while within group T the last three times (T12, T18 and T24) were higher than all preceding them. (*p* = 0.013). There was a difference in anion gap between groups NT and T only at time T24 (*p* = 0.024).

### 3.4. Clinical Variables

Heart rate was higher within the NT group at time T24 than at times T-72 to T6 (*p* = 0.024), while within group T both T24 and T18 were higher than T-72 to T6 (*p* = 0.014). There was no difference between the groups in this regard. Regarding the respiratory rate within both the NT and T groups, higher mean values were found at T6 compared to T18 (*p* = 0.034). There was no difference between NT and T groups regarding respiratory rate. Rectal temperature did not differ between the groups. However, within each group, there was a significant increase in this variable at T6 and T12 after induction. Ruminal movement still existed in about 60% of cattle in both groups until T6, but from this moment until the end of T24 such movements did not exist (Table 3).

Diarrhea, in heavy discharges, started at T6 in about 40% of the animals, reaching the majority of them at T12 and T18, then decreasing its frequency at T24. The presence of ruminitis, enophthalmia, and abdominal distension of the left flank was higher in the treated group (*p* < 0.001) than in the non-treated group, and all clinical variables studied appeared similarly in both groups (Table 4).

### 3.5. Occurrence of Laminitis

Of the 29 animals used in the present study, 27 (93.1%) showed hoof sensitivity. The adapted model with a reduction of the oligofructose doses was efficient for inducing laminitis in most of the animals with no severe ruminal acidosis.

Histological analysis confirmed the occurrence of laminitis characterized by dilation of lymphatic vessels due to dermal edema, vacuolation of epidermal cells, degeneration, and epidermal necrosis, as well as detachment of dermal and epidermal lamellae, basement membrane ripple, capillary proliferation, and moderate lymphocytic infiltrate (Figure 1).

## 4. Discussion

To perform this work, it was necessary to adapt the experimental model to induce laminitis proposed by Thoefner et al. [19], who used dairy bull taurine cows, while in the present work Nelore cows were used. This result confirms the results of comparative studies by Ortolani et al. [9,20] that induced rumen lactic acidosis in taurine and zebuine cattle and found that although ruminal acidosis was identical in them, the resulting clinical picture was different, with marked dehydration in zebu and more pronounced systemic acidosis in taurine cattle.

The initial severe ARA condition indicated the need to decrease the dose of oligofructose employed, with a 10% reduction (15.3 g/kg P.V). As the clinical picture was milder in the first cows tested, later causing the desired laminitis, this sugar dose was maintained. However, some animals later experienced intense clinical manifestations that required one or two additional supportive treatments to prevent death. Therefore, for future use of this model in zebu, a new pilot study with a greater reduction of oligofructose dose would be advisable to ensure a less pronounced clinical picture accompanying the appearance of laminitis. However, it should be noted that Concha et al. [25], employing 13 g oligofructose for Holstein cows, was able to cause ruminal acidosis, but the animals did not show laminitis, unlike for Bustamante et al. [26], who were able to reproduce lameness using the same dose.

As previously mentioned, some of the induced cows (n = 13) required systemic treatment, which could modify the results linked to blood metabolite, blood gas and hydration measurements. Thus, it was decided to divide the groups into treated and non-treated. The moment of greatest ruminal fermentation of oligofructose was until 12 h, identical to that described by Thoefener et al. [19]. From that moment on the ruminal pH increased, and was higher at T24. Although the behavior of the ruminal pH curve between the present work and that of Thoefner et al. [19] is similar, the lowest average value of this variable obtained was 4.4, while in the present work it was 4.02. This reinforces the statement by Ortolani et al. [9] that zebu cattle would be more predisposed to ruminal acidosis than taurine.

The behavior of ruminal pH was not different between cows treated or not, however, the blood pH was much lower in treated animals, which presented systemic acidosis. The main condition that favors the rumen absorption of lactic acid is the presence of ruminal movement during acidosis [27]. However, in the present experiment, all induced animals presented ruminal atony at 12 h and at 6 h about 60% of both groups had ruminal movements. Thus, other factors must positively interfere with the absorption of rumen organic acids, such as their high concentration, low pH and normal osmolarity [8]. The comparison of ruminal osmolarity among cattle treated or not at T6 indicates that it was lower in treated ones, and closer to normal ruminal osmolarity (240 to 300 mOsm/L) [8], facilitating the absorption of ruminal acids in this group. Strengthening this hypothesis, it was found that the difference in blood osmolarity between baseline and 12 h (the beginning of the treatment for most animals) was negatively correlated with blood pH only in the treated group.

Although there was a passage of rumen acids to the blood in the treated animals, a loss of blood fluids to the gastrointestinal system was pronounced at 12h, evidenced by the remarkable dehydration in treated cattle. More than half (8/14) of the untreated cows had had diarrhea at T6h, including diarrheal stool in the stalls, a fact not so evident in the treated animals. The early presence of diarrhea in cattle with ruminal acidosis is a good prognostic sign since it is indicative that the liquids that were sequestered in the rumen left the organ and allowed some of the lost fluids to be reabsorbed in the intestines, mitigating the effects of dehydration [9].

Undoubtedly, in both groups, there was a passage of fluid from the blood to the rumen, evidenced by the abnormal presence of ruminal fluid above the middle third of the left paralumbar fossa that spontaneously spilled this content when the cannula was opened to collect material. At T6 there was already an increase in ruminal content osmolarity in the untreated group, and, to a lesser extent, in the treated animals. Studies using high doses of oligofructose to induce laminitis focused more on systemic changes and laminitis than changes in the rumen environment [19,25,28,29,30]. The present work raised other elements of rumen fermentation. It confirmed the results of Concha et al. (2014) that lactic acid seems to be the main generator of ruminal acidosis. However, in the present work, the coefficient of determination between L rumen lactic acid and pH (*r*^2^ = 0.25) and titratable acidity (*r*^2^ = 0.17) were quite low, suggesting that other acids would also be responsible for acidity in this environment. Although lactic acid plays a major role in the generation of rumen lactic acidosis, short-chain fatty acids also represent an important component in the composition of rumen accumulated acids [8]. According to this author, in lactic acidosis the amount of lactate D corresponds to about 38% of the total lactate produced.

With the fall in rumen pH due to acidosis, several changes in the rumen microbiota occur through the near disappearance of a range of Gram-negative bacteria, including oxygen-reducing and lactate-consuming bacteria, and exponential multiplication of Gram-positive bacteria of lactic acid [1]. The near disappearance of oxygen-reducing bacteria causes this gas to accumulate in the rumen. Two tests performed in this experiment indirectly measured both oxygen accumulation and near disappearance of reducing bacteria; ORP and MBRT.

Both ORP and MBRT results identified a significant increase in these variables during acidosis, especially at T12 and T18, at the peak of the process, decreasing somewhat at T24. The ORP measures the electro voltage within the medium and the higher and positive this value, the higher the degree of oxidability and oxygen content in the rumen medium [31]. MBRT measures the ability of oxygen-consuming bacteria to reduce methylene blue, presented in its oxidized form; a short MBRT indicate normal condition and a higher MBRT means reduced number of bacteria or lower activity [31]. The correlation between ORP and MBRT in the present study was high (*r* = 0.696) indicating the good relationship between these tests. Thus, as might be expected, oligofructose ruminal acidosis causes serious changes in the ruminal environment, similar to what happens in acidosis with other more traditional soluble carbohydrates [32].

There was a clear leukopenia in the animals at T18 and T24, particularly in the treated cattle. Coincidentally, at these times there was an increase in cortisol levels and certainly indications of a super acute inflammatory reaction by both ruminitis, observed by inspection, and the presence of laminitis, from T18 in some cases, but duly apparent at T24h in most animals. Two elements, not analyzed, could cause leukopenia: lymphocytes and neutrophils. It is recognized that a stimulus in cortisol secretions or applications of their analogues causes a decrease in lymphocyte numbers due to the lower proliferative capacity of these cells. But it is also recognized that super acute inflammations cause temporary neutropenia for about 48 h, due to suppression in granulopoiesis or reduction due to high demand [33].

Oligofructose acidosis certainly caused inflammation and pain in the animals, because in most of them ruminitis and later laminitis were found in the vast majority, which was confirmed by the histopathological results. Cortisol is considered a specific stress marker that may be due to an inflammatory process, intense pain, excessive manipulation, or a threatening situation to the animal [34]. Even before laminitis developed, females had a large increase in cortisol concentration, increasing fivefold over the first 24 h of induction. Both treated and non-treated animals showed similar increases in cortisol concentrations, indicating that there was no marked influence of dehydration, metabolic acidosis, or extra manipulation to perform the treatment.

The present results can be compared to those obtained by Bustamante et al. [26], who induced acidosis in Holstein heifers with 13 g/kg oligofructose, a slightly lower amount of oligofructose than this study, and found that cortisol concentration doubled during the 24th h of induction. The higher cortisol increasing in Nelore cattle may be associated with the animal temperament.

The animals treated have been shown to have higher dehydration, already discussed, as well as a more prominent metabolic acidosis compared to untreated cattle. The degree of acidemia in the medicaments was considered relatively severe. Also, an indicator of metabolic acidosis was the anion gap that increased, reaching maximum values at T18 and T24. This was due to the decrease in bicarbonate levels caused by metabolic acidosis, accompanied by the maintenance of sodium and potassium concentrations.

Dehydration was probably the expected cause of the increased creatinine and serum urea concentration at T24 in treated animals, which presented higher levels than untreated cattle. Classically, dehydration causes less renal blood flow and consequently less filtration of these metabolites increasing their blood concentrations [22].

Dehydration was more intense in treated animals in which symptoms such as dry muffle, enophthalmos and cold extremities were detected, indicating increased degrees of dehydration [35]. Nasal secretion appeared abundantly with seromucous or catarrhal mucus characteristics, and may appear in one or two nostrils, usually at T12 in most animals.

Regarding blood gas variables, the decrease in blood pCO_2_ at the peak of metabolic acidosis is noteworthy, interpreted as a clear respiratory compensation to metabolic acidosis, which was also observed by Noronha Filho et al. [36]. 

The clinical picture during acidosis induction was the expected and classic described in cattle with severe rumen lactic acidosis. Thus, in both treated and untreated cows, there was a significant tachycardia, especially when dehydration and systemic acidosis arose, which triggered a sympathetic stimulus that generates the phenomenon [9]. Tachycardia has possibly also influenced a reduction in blood pressure present in hypovolemia, leading to a compensatory response to sinus tachycardia; such a condition has already been described in cattle with rumen lactic acidosis [9].

Another very evident sign was ataxia, present in all cases at T12, in the 27 animals of the study. Although not discriminated in the results, females with greater depression and more intense dehydration and systemic acidosis presented more pronounced ataxia. This ataxia could be a consequence of laminitis, but the other two cattle discarded from the experiment, due to the absence of foot problems, also showed a reluctance to walk. This symptom is quite common in cattle with sucrose-induced rumen lactic acidosis, and most of them do not have laminitis, which leads us to believe that this ataxia is directly related to the deleterious systemic effects of rumen acidosis itself.

An interesting fact observed in the induction with oligofructose was the hyperthermia detection at T6, reaching its peak at T12, in both treated and untreated animals. As this hyperthermia was accompanied by tachypnea, tachycardia, and ocular mucosal congestion, it is suggested that these manifestations characterize fever syndrome.

Considering the normal range of rectal temperature of beef cattle, which is between 38.1 °C and 39.1 °C, and the maximum average values obtained at T12 (39.6 °C) we can infer that induced cattle had a mild transient fever, since in the T18 ha temperature dropped [37]. Such clinical description was similarly found by Lohuis et al. [38], who injected different amounts of bacterial lipopolysaccharides (LPS) into adult cattle and found hyperthermia from the 3rd h of inoculation, without returning to normal temperature within eight hours.

It is interesting to compare the clinical picture between the induction of lactic acidosis with sucrose or oligofructose in cattle. In the latter, no hyperthermia was described during the process, in contrast to that observed in a study with sucrose [24]. It is noteworthy that the appearance of laminitis after the induction of acidosis with sucrose was rare, which was the opposite when oligofructose was used. Thus, due to the clinical findings of short-term hyperthermia and high frequency of laminitis, it is strongly assumed that in the induction with oligofructose there is a high generation of rumen LPS with the absorption of these compounds, leading to the onset of foot problem.

Ruminitis, found by inspection, was more often present in treated than untreated animals. Ruminitis was found at T24, when the ruminal fluid content was no longer pronounced and the abdomen was not so distended, reaching its maximum at T12. According to Owens et al. [8] ruminitis may be derived from the pronounced drop in pH, an exaggerated increase in osmolarity of ruminal content and exacerbated passage of blood fluids to the rumen, causing swelling in the rumen papillae cells. Although there were no striking differences between pH and ruminal osmolarity between the treated and untreated groups, there was undoubtedly a greater process of passage of blood fluids to the rumen in treated cattle, evidenced by the greater number of bovines with distension on the left flank. The present study was successful in inducing laminitis in 93.1% of zebu animals, while Thoefner et al. [19] reached 66.7% and Danscher et al. [39] reached 100% in taurine cows. The histological findings are compatible with the inflammatory process, confirming the occurrence of laminitis, with similar features to those found by others Noronha Filho et al. [36] and Danscher et al. [28], with biopsies performed at 28 and 72 h, respectively.

## 5. Conclusions

The amount of oligofructose required to induce laminitis in zebu had to be decreased compared to the classic taurine protocol, due to severe ruminal and systemic acidosis caused by the original dose, which promoted a prominent ARA, decrease in rumen pH, accumulation of lactic acid, intense decrease of anaerobiosis, and temporary increase in ruminal osmolarity, and required the use of treatment to correct intense metabolic acidosis and dehydration.

The clinical picture caused by excess oligofructose using the reduced doses is more severe, with the differential of the appearance of an ephemeral fever, respiratory compensation in the systemic acidosis, and laminitis.

## Figures and Tables

**Figure 1 animals-10-00429-f001:**
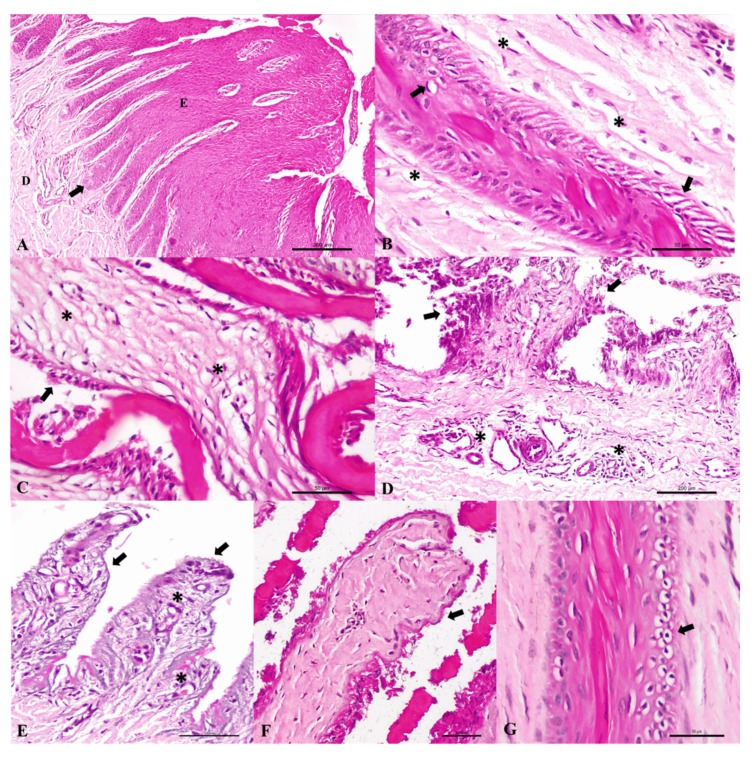
Histopathologic finding observed in zebu cattle maintained on oligofructose. (**A**) Horizontal claw section of the lamellar region from control animal. The normal dermo-epidermal junction consists of numerous interlocking dermal (**D**) and epidermal (**E**) lamellae. The tips of the epidermal lamellae (arrow) are normally smooth and rounded. (**B**–**G**) Cross-section of the lamellar region from zebu cattle given an alimentary oligofructose overload. (**B**) There is dilation of lymphatic vessels by dermal edema (*) and epidermal degeneration (arrows). (**C**) Necrosis epidermal (arrow) and severe dermal edema (*). (**D**) Detachment and necrosis of epidermal lamellae (arrows) and moderate lymphocytic dermal infiltrate (asterisks). (**E**) Severe epidermal necrosis (arrows) and proliferation of capillaries (*). (**F**) In the epidermal lamellae observed necrosis and basement membrane ripple (arrow) and dermal fibrosis. (**G**) Moderate vacuolar degeneration in epidermal lamellae. Hematoxylin and Eosin stain. Bar, A and D 200 μm; B, C, E–G 50 μm.

**Table 1 animals-10-00429-t001:** Mean and standard deviation of ruminal (pH, ORP, AcT and MBRT) and blood (lactate L, osmolarity and cortisol) variables of cattle submitted to acute rumen lactic acidosis induction by oligofructose overload.

Time	Variables						
	Ruminal Fluid				Blood		
	pH	ORP (mV)	TA (mL)	MBRT (min)	L-Lactate (mmol/L)	Osmolarity (mOsm/L)	Cortisol (nmol/L)
T-72	6.50 ± 0.3 ^a^	36.8 ± 20.7 ^d^	3.13 ± 0.8 ^d^	1.07 ± 0.7 ^c^	1.07 ± 0.3 ^ab^	302.89 ± 18.7 ^b^	55.7 ± 33.1 ^c^
T-48	6.36 ± 0.3 ^a^	43.2 ± 18.0 ^d^	3.27 ± 0.9 ^d^	0.56 ± 0.39 ^c^	0.96 ± 0.4 ^ab^	303.00 ± 19.3 ^b^	49.1 ± 24.8 ^c^
T-24	6.47 ± 0.2 ^a^	36.0 ± 17.7 ^d^	3.21 ± 0.9 ^d^	0.41 ± 0.26 ^c^	0.83 ± 0.3 ^b^	305.25 ± 23.4 ^b^	44.4 ± 19.3 ^c^
T0	6.37 ± 0.1 ^a^	41.7 ± 10.0 ^d^	3.38 ± 0.9 ^d^	0.51 ± 0.29 ^c^	0.78 ± 0.2 ^b^	312.33 ± 12.4 ^b^	53.8 ± 33.1 ^c^
T6	4.67 ± 0.2 ^b^	144.4 ± 10.3 ^c^	5.77 ± 1.8 ^c^	20.72 ± 13.2 ^b^	0.79 ± 0.2 ^b^	330.82 ± 25.2 ^a^	84.7 ± 46.9 ^c^
T12	4.12 ± 0.1 ^cd^	175.8 ± 7.1 ^a^	8.08 ± 1.4 ^ab^	52.43 ± 13.6 ^a^	1.01 ± 0.3 ^ab^	316.86 ± 25.9 ^ab^	198.9 ± 63.4 ^b^
T18	4.02 ± 0.2 ^d^	181.4 ± 12.3 ^a^	8.58 ± 2.3 ^a^	49.87 ± 18.3 ^a^	0.98 ± 0.3 ^ab^	316.59 ± 24.2 ^ab^	274.7 ± 115.8 ^a^
T24	4.34 ± 0.4 ^c^	160.5 ± 27.9 ^b^	7.14 ± 1.5 ^b^	30.18 ± 23.0 ^b^	1.24 ± 0.5 ^a^	312.60 ± 22.1 ^ab^	275.6 ± 113.1 ^a^

Different lowercase letters in the same column mean difference between moments. ORP: oxide-reduction potential; TA: titratable acidity; MBRT: methylene blue reduction time (MBRT).

**Table 2 animals-10-00429-t002:** Mean and standard deviation of hematological parameters: globular volume (VG), hemoglobin, red blood cell count (RBC) and white blood cell count (WBC) of cattle submitted to acute rumen lactic acidosis induction in treated (T) and non-treated (NT) animals.

Variables		Times							
		T-72	T-48	T-24	T0	T6	T12	T18	T24
Hematocrit (%)	NT	31.9 ± 2.7 ^b^	32.8 ± 3.6 ^b^	32.3 ± 2.3 ^b^	31.5 ± 2.7 ^b^	32.0 ± 1.6 ^Bb^	34.3 ± 3.8 ^Bb^	41.1 ± 5.5 ^Ba^	41.0 ± 5.3 ^a^
	T	34.5 ± 5.5 ^cd^	34.3 ± 3.7 ^cd^	33.8 ± 2.6 ^d^	33.9 ± 3.6 ^d^	35.3 ± 3.0 ^Acd^	39.9 ± 5.4 ^Abc^	46.2 ± 6.4 ^Aa^	44.1 ± 6.0 ^ab^
RBC (×10^6^/mm^3^)	NT	7.25 ± 0.7 ^b^	7.42 ± 0.8 ^b^	7.44 ± 0.5 ^b^	7.17 ± 0.5 ^b^	7.25 ± 0.4 ^Bb^	7.74 ± 0.9 ^Bb^	9.19 ± 1.2 ^Ba^	9.12 ± 1.2 ^a^
	T	7.85 ± 1.0 ^cd^	7.83 ± 0.7 ^d^	7.73 ± 0.7 ^d^	7.79 ± 0.8 ^d^	8.02 ± 0.7 ^Acd^	9.04 ± 0.9 ^Abc^	10.29 ± 1.2 ^Aa^	9.96 ± 9.5 ^ab^
Hemoglobin (g/dL)	NT	10.2 ± 0.9 ^b^	10.4 ± 0.9 ^b^	10.2 ± 0.6 ^b^	9.9 ± 0.8 ^b^	9.9 ± 0.6 ^Bb^	10.6 ± 1.2 ^Bb^	13.1 ± 2.5 ^Ba^	12.5 ± 1.6 ^a^
	T	11.0 ± 1.5 ^cd^	10.9 ± 0.9 ^cd^	10.6 ± 0.9 ^d^	10.9 ± 1.1 ^cd^	11.2 ± 0.9 ^Acd^	12.4 ± 1.4 ^Abc^	14.2 ± 1.8 ^Aa^	13.8 ± 1.7 ^ab^
WBC (×10^3^/mm^3^)	NT	10.0 ± 2.1	10.2 ± 2.5	9.9 ± 1.8	10.2 ± 2.4	11.1 ± 2.4	10.6 ± 4.3	7.8 ± 2.4	8.8 ± 2.0
	T	12.2 ± 2.7 ^a^	11.6 ± 2.0 ^ab^	11.8 ± 1.5 ^ab^	11.3 ± 1.8 ^abc^	11.3 ± 1.7 ^abc^	12.4 ± 2.1 ^a^	8.9 ± 2.5 ^bc^	8.6 ± 2.5 ^c^

Different lowercase letters in the same line mean difference between moments. Different capitalized letter stands for difference between groups (*p* < 0.05). NT: non-treated, T: Treated, RBC: red blood cell; WBC: white blood cell.

**Table 3 animals-10-00429-t003:** Mean and standard deviation of ruminal (L-lactate and osmolarity), blood gas (pH, bicarbonate and base excess (BE)), biochemical (Urea and creatinine) and clinical (variables of cattle submitted to acute rumen lactic acidosis induction in treated (T) and non-treated (NT) animals.

Variables		Times							
		T-72	T-48	T-24	T0	T6	T12	T18	T24
L-Lactate (mmol/L)	NT	0.02 ± 0.01 ^c^	0.02 ± 0.02 ^c^	0.03 ± 0.07 ^c^	0.01 ± 0.01 ^c^	40.0 ± 8.3 ^b^	57.4 ± 8.9 ^a^	51.0 ± 9.6 ^a^	39.3 ± 15.3 ^b^
	T	0.02 ± 0.03 ^b^	0.07 ± 0.09 ^b^	0.10 ± 0.1 ^b^	0.02 ± 0.03 ^b^	42.2 ± 8.9 ^ab^	51.8 ± 9.1 ^a^	53.2 ± 9.9 ^a^	39.6 ± 16.9 ^b^
Osmolarity (mOsm/L)	NT	289.1 ± 33 ^ab^	262.3 ± 29 ^b^	283.6 ± 51 ^b^	256.6 ± 39 ^b^	336.0 ± 22 ^Aa^	288.3 ± 38 ^ab^	242.8 ± 37 ^Bb^	269.0 ± 35 ^b^
	T	285.2 ± 41	287.6 ± 36	296.7 ± 46	263.3 ± 51	306.6 ± 37 ^B^	268.2 ± 35	270.6 ± 34 ^A^	266.9 ± 45
pH	NT	7.44 ± 0.03 ^a^	7.44 ± 0.04 ^a^	7.42 ± 0.03 ^a^	7.43 ± 0.04 ^a^	7.41 ± 0.03 ^Aa^	7.27 ± 0.04 ^Ab^	7.29 ± 0.07 ^Ab^	7.33 ± 0.08 ^b^
	T	7.41 ± 0.04 ^a^	7.39 ± 0.06 ^a^	7.38 ± 0.06 ^a^	7.41 ± 0.03 ^a^	7.34 ± 0.1 ^Bab^	7.18 ± 0.07 ^Bc^	7.19 ± 0.08 ^Bc^	7.29 ± 0.07 ^b^
Bicarbonate (mmol/L)	NT	26.3 ± 2.7 ^a^	25.9 ± 2.6 ^a^	25.1 ± 1.9 ^a^	25.1 ± 1.9 ^Aa^	24.6 ± 3.7 ^a^	14.2 ± 2.3 ^Ab^	12.8 ± 4.0 ^Ab^	15.6 ± 5.3 ^Ab^
	T	23.7 ± 3.2 ^a^	23.8 ± 3.2 ^a^	22.8 ± 2.7 ^a^	23.3 ± 2.9 ^Ba^	22.7 ± 2.9 ^a^	11.1 ± 1.6 ^Bb^	8.7 ± 2.5 ^Bb^	11.0 ± 3.2 ^Bb^
BE (mmol/L)	NT	2.8 ± 2.8 ^Aa^	2.0 ± 3.0 ^Aa^	1.0 ± 2.3 ^Aa^	1.2 ± 2.2 ^Aa^	0.6 ± 4.2 ^a^	−12.0 ± 2.8 ^Ab^	−13.2 ± 5.0 ^Ab^	−9.7 ± 6.6 ^Ab^
	T	0.45 ± 3.4 ^Ba^	0.84 ± 2.8 ^Ba^	−1.69 ± 2.9 ^Ba^	−0.92 ± 2.7 ^Ba^	−1.61 ± 2.9 ^a^	−15.4 ± 2.9 ^Ba^	−19.5 ± 3.4 ^Bb^	−15.9 ± 4.4 ^Bb^
Urea (mmol/L)	NT	4.19 ± 1.7 ^a^	3.36 ± 1.2 ^ab^	3.38 ± 1.4 ^ab^	2.98 ± 1.1 ^ab^	2.22 ± 1.1 ^Bb^	1.80 ± 0.7^b^	2.17 ± 0.8 ^b^	2.60 ± 1.1 ^Bab^
	T	4.51 ± 1.4 ^a^	3.35 ± 1.0 ^abc^	3.13 ± 0.9 ^bc^	3.68 ± 1.0 ^ab^	2.73 ± 0.8 ^Abc^	2.29 ± 0.6 ^c^	2.34 ± 0.7 ^c^	3.46 ± 0.9 ^Aabc^
Creatinine (mg/dL)	NT	1.73 ± 0.2	1.74 ± 0.2	1.82 ± 0.3	1.76 ± 0.1	1.93 ± 0.1	1.93 ± 0.2	1.98 ± 0.4	1.95 ± 0.3 ^B^
	T	1.76 ± 0.3 ^b^	1.76 ± 0.2 ^b^	1.73 ± 0.2 ^b^	1.78 ± 0.2 ^b^	1.95 ± 0.2 ^b^	2.02 ± 0.3 ^ab^	2.15 ± 0.2 ^ab^	2.50 ± 0.5 ^Aa^
HR (beats/min)	NT	74.4 ± 9.8 ^cd^	72.4 ± 13.2 ^d^	71.4 ± 9.0 ^d^	74.3 ± 10.6 ^cd^	83.2 ± 13.6 ^bcd^	89.0 ± 13.3 ^ab^	97.6 ± 16.1 ^ab^	105.2 ± 8.8 ^a^
	T	64.0 ± 8.7 ^d^	69.5 ± 9.7 ^cd^	71.8 ± 10.6 ^cd^	74.7 ± 13.8 ^cd^	82.7 ± 9.9 ^bc^	94.6 ± 22.7 ^ab^	107.2 ± 15.8 ^a^	109.1 ± 9.1 ^a^
RR (mov/min)	NT	24.7 ± 8.0 ^ab^	24.1 ± 4.7 ^ab^	24.3 ± 5.0 ^ab^	26.1 ± 5.8 ^ab^	32.0 ± 8.6 ^a^	23.6 ± 5.1 ^ab^	22.7 ± 8.7 ^b^	24.1 ± 5.5 ^ab^
	T	21.1 ± 6.4 ^ab^	22.4 ± 5.0 ^ab^	22.2 ± 4.2 ^ab^	24.9 ± 5.6 ^ab^	28.6 ± 8.6 ^a^	27.6 ± 5.2 ^ab^	20.8 ± 3.6 ^b^	21.3 ± 4.6 ^ab^
RT (C)	NT	38.4 ± 0.2^b^	38.4 ± 0.2^b^	38.6 ± 0.3 ^b^	38.6 ± 0.2 ^b^	39.1 ± 0.4 ^a^	39.6 ± 0.5 ^a^	38.5 ± 0.6 ^b^	38.8 ± 0.5 ^b^
	T	38.6 ± 0.2^b^	38.5 ± 0.1^b^	38.3 ± 0.2 ^b^	38.6 ± 0.2 ^b^	39.4 ± 0.2 ^a^	39.6 ± 0.4 ^a^	38.7 ± 0.8 ^b^	38.5 ± 0.9 ^b^

Different lowercase letters in the same line mean difference between moments. Different capitalized letter stands for difference between groups (*p* < 0.05). NT: non-treated, T: Treated, BE: base excess, HR: heart rate, RR: respiratory rate, RT: rectal temperature.

**Table 4 animals-10-00429-t004:** Clinical manifestations of cattle submitted to acute rumen lactic acidosis induction in the medicated (M) and non-medicated (NT) groups.

Clinical Manifestations	Groups	
	NT	T
Nasal discharge	11/14	13/13
Epiphora	1/14	1/13
Dry muffle	11/14	13/13
Enophthalmos	4/14 ^B^	9/13 ^A^
Cold extremities	0/14	3/13
Muscular tremors	2/14	3/13
Ataxia at T12	14/14	13/13
Ruminitis at T24	4/14 ^B^	9/13 ^A^
Left flank distension at T12	1/7 ^B^	7/7 ^A^

Different capitalized letter stands for difference between groups (*p* < 0.05). NT: non-treated, T: Treated.

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
