# Peer review of "Characterization of Oligofructose-Induced Acute Rumen Lactic Acidosis and the Appearance of Laminitis in Zebu Cattle"

_animals, 2020, doi:10.3390/ani10030429_

Round 1
Reviewer 1 Report
Introduction and discussion section should be improved: Lines 62-64: This sentence is imprecise/uncorrect. Ruminal and metabolic acidosis increase ROS production, and this oxidative stress may prevent all the compensatory mechanisms to alleviate the problem. In fact, H+ difussion from the rumen to the blood reduces blood pH values, thus promoting metabolic acidosis. This causes the reduciton of Anion Gap (you can mention this effect on line 453). Moreover, metabolic acidosis increases H+ leakage to the mitochondria, thus inhibiting ATP production. Piruvate is diverted towards lactate production, CO2 is increased and therefore hypercapnia appears. Under these circumstances hyperventilation should take place to compensate hypercapnia, but the binding tenacity of Hb by O2 is increased to reduce ROS damage in cells. Therefore cells keep producing acid lactic with no recovery (lactic acid trap). This could be expalined in Line 450, where the authors mention "unknown reason".
Author Response
We appreciate the reviewer for the contribution to our work, which certainly has improved.
Regarding the specific comments:
Introduction and discussion section should be improved: Lines 62-64: This sentence is imprecise/uncorrect. Ruminal and metabolic acidosis increase ROS production, and this oxidative stress may prevent all the compensatory mechanisms to alleviate the problem. In fact, H+ difussion from the rumen to the blood reduces blood pH values, thus promoting metabolic acidosis. This causes the reduciton of Anion Gap (you can mention this effect on line 453). Moreover, metabolic acidosis increases H+ leakage to the mitochondria, thus inhibiting ATP production. Piruvate is diverted towards lactate production, CO2 is increased and therefore hypercapnia appears. Under these circumstances hyperventilation should take place to compensate hypercapnia, but the binding tenacity of Hb by O2 is increased to reduce ROS damage in cells. Therefore cells keep producing acid lactic with no recovery (lactic acid trap). This could be expalined in Line 450, where the authors mention "unknown reason".
In fact, our sentence was: Metabolic acidosis may cause oxidative stress, as mitochondria are stimulated to increase reactive oxygen species (ROS) production and to decrease adenosine triphosphate (ATP) production, which may further aggravate the condition of the animal.
Thus, our sentence in the introduction is in accordance with the comment made by the reviewer. The increase of ROS, decrease of ATP, which aggravate the condition of the animals.
Regarding the discussion, we changed accordingly. Please see the revised manuscript. The text mentioned (unknown reason) was only to cite that the compensatory hyperventilation did not occur in another experiment. But we check further studies and in some, they occur, thus we revised this part of the discussion and move a few paragraphs for better readability.
Additionally, we correct the manuscript for typos and for standardization.
Reviewer 2 Report
All my comments were corrected appropriately, i recommend to accept the new version of the manuscript
Author Response
We appreciate the reviewer for the contribution to our work, which certainly was improved.
This manuscript is a resubmission of an earlier submission. The following is a list of the peer review reports and author responses from that submission.
Round 1
Reviewer 1 Report
xxv
Title: Characterization of oligofructose-induced acute rumen lactic
acidosis and the appearance of laminitis in Zebu cattle
Journal: Animals
The manuscript is very well written, with a clear description of material and methods, a proper experimental design, and correct statistics. However, the manuscript is too long and an effort of the authors should be done before considering this manuscript to be accepted in Animals. In my opinion, the part describing the results can be considerably shortened.
Other minor details
How did the authors assay leukopenia? Hematology profile is not assayed in the tables.
Please, define the abbreviations in tables.
Line 383. Please, change “flora” into “microbiota”.
Please, include references of other mechanisms related to ruminal and systemic acidosis establishment. Recently an involvement of oxidative stress has been highlighted in several publications such as:
Mobbs, C., 2007. Oxidative stress and acidosis, molecular responses to, in: Finck, G., (Ed.), Encyclopedia of Stress. Academic Press, San Diego, CA, p. 49.
Metabolic acidosis corrected by including antioxidants in diets of fattening lambs. Small Ruminant Research 109(s 2–3):133–135. DOI: 10.1016/j.smallrumres.2012.08.009
Author Response
The manuscript is very well written, with a clear description of material and methods, a proper experimental design, and correct statistics. However, the manuscript is too long and an effort of the authors should be done before considering this manuscript to be accepted in Animals. In my opinion, the part describing the results can be considerably shortened.
Answer: We remove some parts reducing the original manuscript, removing one table and few paragraphs, but according to the other reviewer suggestion we add more data and include a new figure. Since the journal does not have page limit, we believe that the length will not be a limitation to publish.
Other minor details
How did the authors assay leukopenia? Hematology profile is not assayed in the tables.
A: The hematology data was added as requested.
Please, define the abbreviations in tables.
A: The correction was made as requested.
Line 383. Please, change “flora” into “microbiota”.
A: The correction was made as requested.
Please, include references of other mechanisms related to ruminal and systemic acidosis establishment. Recently an involvement of oxidative stress has been highlighted in several publications such as:
A: references were added.
Reviewer 2 Report
The manuscript assess the effects of oligofructose on acute ruminal acidosis induction and the appearance of laminitis in Zebu cattle.
Has been widely described before in cattle that oligofructose induces acute ruminal acidosis, therefore information about potential breed differences should be included in the introduction section. Why is necessary to perform this experimental procedure in zebu cattle?
My main concern about the manuscript is the description of laminitis.
Since is the first use of oligofructose in zebu cattle, the appearance of laminitis should be confirmed by histological analysis. Other authors describe lameness as clinical findings during oligofructose overload, however in the present article did not show evidence that the hoof sensitivity test is associated with alterations in animals locomotion.
Line 54: the authors describe: “acute rumen acidosis (ALRA)…” do you mean ARA?
line 72 …“Laminite”… please correct
The authors describe the statistical criteria for significance as p=0.05, please correct for p<0.05
The results of urea and creatinine should be included in table 3
Table 3 the authors described the groups T (treated) and NT (non treated) however is showed “M” group as non treated, please review. In addition, the term treated (T) is some confuse in text since also is used to described time period. e.g. Line 286-290 the authors describe 12h, 18h or 24h however in other part of the text T12, T18, or T24, please use a similar form to describe time period in whole text.
Line 130 a reference is missing
The authors describe some parameters in material and methods section, but in results is shown RBC, He? VG, please review carefully and described appropriately. In addition the authors mentioning that this parameters are described in table 3 line 269-270 but were not included.
Line 286, authors use BE term, this acronyms means “base excess”?
In table 3, the lactate concentrations in NT group at T12 was 57.4±81.9 mmol/L the standard deviation is correct?. T6 and T18 in the same group show lesser values of SD.
Line 300-307 the physiological parameters values should be included in table
The authors mentioning laminitis according to hoof sensitivity, however did not show histological evidence to sustain this conclusion
Line 337, Bustamante et al., described the presence of lameness according to locomotion score, but not laminitis, as was suggested in discussion.
In discussion section should be avoid to describe p values.
Line 350-351 the correlation of decrease of blood pH and ruminal pH “could” indicate a possible passage of acid from rumen to blood. However, only L-lactate was measured and not D-lactate or short fatty acids, which are produced mainly by ruminal bacteria. Line 361, the experiments performed did not include the measuring of organic acid absorption from rumen, therefore this conclusion is not supported with an appropriate experimental design.
Line 401, authors mentioning “super acute inflammation reaction” according to ruminitis (by visual inspection) and laminitis, however not inflammatory markers (e.g. haptoglobin) or histological analysis were included in order to confirm this suggestion
Line 458, review the sentence.
Line 466-468, the sentence is too speculative
Author Response
The manuscript assess the effects of oligofructose on acute ruminal acidosis induction and the appearance of laminitis in Zebu cattle. Has been widely described before in cattle that oligofructose induces acute ruminal acidosis, therefore information about potential breed differences should be included in the introduction section.
A: the correction was made, please see lines 78-79.
Why is necessary to perform this experimental procedure in zebu cattle? My main concern about the manuscript is the description of laminitis. Since is the first use of oligofructose in zebu cattle, the appearance of laminitis should be confirmed by histological analysis. Other authors describe lameness as clinical findings during oligofructose overload, however in the present article did not show evidence that the hoof sensitivity test is associated with alterations in animals locomotion.
A: The study is necessary since the main breed from Brazil used in confinement is Nelore and there is no report about this model in such breed. To evaluate future protocols of treatments of laminitis using experimental models our adapted methodology should be followed. The Zebu respond differently than Taurine cattle. As requested, we add the histologic analysis that confirmed the laminitis. Please see the new figure 1. This manuscript focuses on the physical, blood and ruminal aspects of Nelore cattle induced. We submit another manuscript regarding the diagnostic tests to the Journal of Animal Science. This second manuscript was requested major revision and now is pending decision. The pain sensitivity was strongly associated with locomotion score.
Line 54: the authors describe: “acute rumen acidosis (ALRA)…” do you mean ARA?
A: Yes. The correction was made throughout the manuscript.
line 72 …“Laminite”… please correct
A: corrected as requested.
The authors describe the statistical criteria for significance as p=0.05, please correct for p<0.05
A: corrected as requested.
The results of urea and creatinine should be included in table 3
A: corrected as requested.
Table 3 the authors described the groups T (treated) and NT (non treated) however is showed “M” group as non treated, please review. In addition, the term treated (T) is some confuse in text since also is used to described time period. e.g. Line 286-290 the authors describe 12h, 18h or 24h however in other part of the text T12, T18, or T24, please use a similar form to describe time period in whole text.
A: corrected as requested.
Line 130 a reference is missing.
A: corrected as requested (the references are not marked because we used software).
The authors describe some parameters in material and methods section, but in results is shown RBC, He? VG, please review carefully and described appropriately. In addition the authors mentioning that this parameters are described in table 3 line 269-270 but were not included.
A: For a better presentation of the data, the original Table 2 was replaced by data on hematological variables (hematocrit, the number of red blood cells, leukocytes and hemoglobin concentration. The methodology section was also corrected.
Line 286, authors use BE term, this acronyms means “base excess”?
A: Yes, BE is base excess. We define the abbreviation in the manuscript.
In table 3, the lactate concentrations in NT group at T12 was 57.4±81.9 mmol/L the standard deviation is correct? T6 and T18 in the same group show lesser values of SD.
A: The SD correct is 8.9, it was a typo. Sorry for this mistake and thank you for the correction.
Line 300-307 the physiological parameters values should be included in table.
A: the value was included, please see the corrected table
The authors mentioning laminitis according to hoof sensitivity, however did not show histological evidence to sustain this conclusion.
A: the histopathological finding were included, please see figure 1.
Line 337, Bustamante et al., described the presence of lameness according to locomotion score, but not laminitis, as was suggested in discussion.
A: corrected as requested.
In discussion section should be avoid to describe p values.
A: corrected as requested.
Line 350-351 the correlation of decrease of blood pH and ruminal pH “could” indicate a possible passage of acid from rumen to blood. However, only L-lactate was measured and not D-lactate or short fatty acids, which are produced mainly by ruminal bacteria. Line 361, the experiments performed did not include the measuring of organic acid absorption from rumen, therefore this conclusion is not supported with an appropriate experimental design.
A: We agree with the reviewer and the sentence was removed.
Line 401, authors mentioning “super acute inflammation reaction” according to ruminitis (by visual inspection) and laminitis, however not inflammatory markers (e.g. haptoglobin) or histological analysis were included in order to confirm this suggestion.
A: the histology data confirmed the occurrence of laminitis.
Line 458, review the sentence.
A: corrected as requested.
Line 466-468, the sentence is too speculative
A: Since the histology data were added, now we can confirm that laminitis occurred.